# Muscle AMP deaminase activity was lower in Neandertals than in modern humans

Dominik Macak [1], Shin-Yu Lee [2], Tomas Nyman [3], Henry Ampah-Korsah[3], Emilia Strandback[3], Svante Pääbo [1,2] & Hugo Zeberg [1,4] ✉

The enzyme AMPD1 is expressed in skeletal muscle and is involved in ATP production. All available Neandertal genomes carry a lysine-to-isoleucine substitution at position 287 in AMPD1. This variant, which occurs at an allele frequency of 0–8% outside Africa, was introduced to modern humans by gene flow from Neandertals. Here, we show that the catalytic activity of the purified Neandertal AMPD1 is ~25% lower than the ancestral enzyme, and when introduced in mice, it reduces AMPD activity in muscle extracts by ~80%. Among present-day Europeans, another AMPD1 variant encoding a stop codon occurs at an allele frequency of 9–14%. Individuals heterozygous for this variant are less likely to be top-performing athletes in various sports, but otherwise reduced AMPD1 activity is well tolerated in present-day humans. While being conserved among vertebrates, AMPD1 seems to have become less functionally important among Neandertals and modern humans.

Purines are essential as DNA and RNA building blocks, brain signaling molecules, and energy carriers such as ATP. Surprisingly, human tissues, particularly the brain, exhibit lower purine levels compared to chimpanzees and macaques[1]. The enzyme adenylosuccinate lyase (ADSL) that catalyzes two reactions in purine de novo synthesis (Fig. 1a), carries an amino acid substitution that is unique to nearly all present-day humans but absent in Neandertals and Denisovans, two 'archaic' human forms that split from modern humans about 600,000 years ago. Introducing the ancestral ADSL into human cells increases purine levels, while the modern human version decreases purine concentrations in mice[1]. Thus, purine biosynthesis has been reduced on the lineage leading to modern humans after their separation from the ancestors shared with archaic humans.

To investigate if purine metabolism may have changed also on the Neandertal lineage, we analyze all available Neandertal genomes with respect to amino acid variants in enzymes involved in purine metabolism. We find an amino acid substitution in the enzyme adenosine monophosphate deaminase 1 (AMPD1) that is present in all Neandertal genomes. AMPD1 is expressed in skeletal muscle and catalyzes the conversion of adenosine monophosphate (AMP) to inosine monophosphate (IMP) and ammonia in the purine nucleotide cycle. This reaction shifts the equilibrium of the myokinase reaction (2 ADP ↔ ATP + AMP) towards the production of ATP and AMP and is considered to be important for normal muscle function[2–6]. We characterize the Neandertal version of AMPD1 and find that it is less active than the ancestral version of the enzyme and show that some present-day humans carry the Neandertal enzyme as a result of gene flow. This, as well as another AMPD1 substitution in present-day humans, suggest that AMPD1 has become less crucial in modern humans and Neandertals.

## Results

### An amino acid substitution in Neandertal AMPD1

To investigate potential genetic differences in purine metabolism between modern humans and Neandertals, we compared inferred human ancestral alleles based on the alignment of eight primate genomes (Ensembl release 84, ref. [7]) with the genomes of three high-coverage Neandertals[8–10] across 128 genes involved in purine metabolism, as annotated in the Kyoto Encyclopedia of Genes and Genomes[11] (KEGG pathway hsa00230; Supplementary Fig. 1). We focused on variants where the ancestral allele is fixed in 661 African genomes[12] and the derived allele is homozygous in the three

[1]Max Planck Institute for Evolutionary Anthropology, Leipzig, Germany. [2]Okinawa Institute of Science and Technology, Onna-son, Japan. [3]Protein Science Facility, Department of Medical Biochemistry and Biophysics, Karolinska Institutet, Stockholm, Sweden. [4]Department of Physiology and Pharmacology, Karolinska Institutet, Stockholm, Sweden. ✉e-mail: hugo.zeberg@ki.se

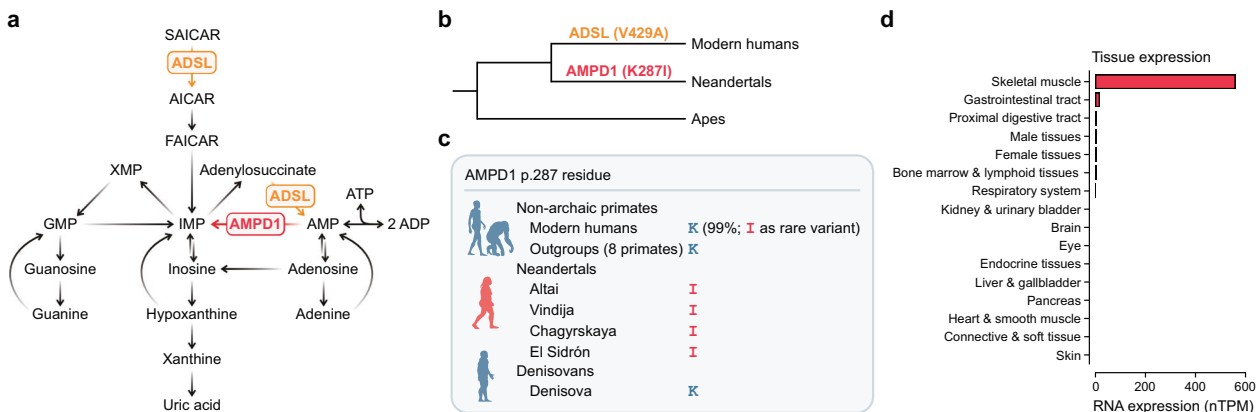

**Fig. 1 | Amino acid differences between modern humans and Neandertals in enzymes involved in purine metabolism. a** The de novo and salvage purine biosynthesis pathway, including the myokinase reaction (2 ADP ↔ ATP + AMP). **b** Amino acid substitutions in purine metabolism enzymes on human lineages. **c** Amino acid at the corresponding position 287 in the protein sequence of AMPD1 of non-archaic primates, Neandertals and Denisovans. **d** Tissue RNA expression of *AMPD1* based on transcriptomic data from a consensus dataset (Human Protein Atlas[88] and Genotype-Tissue Expression project[89]). Skeletal muscle includes tongue. Data shown as normalized number of transcripts per million (nTPM). Source data are provided as a Source Data file.

Neandertal genomes. Applying these filtering criteria, we identified four missense variants in proteins involved in purine metabolism, located in the genes *AMPD1*, *ADPRM*, *PDE1C*, and *PDE4A* (Supplementary Table 1). Notably, *AMPD1* encodes a key enzyme in the purine nucleotide cycle functioning immediately downstream of ADSL (Fig. 1a), and is therefore the focus of this study. The Neandertal-specific missense variant in *AMPD1* (rs34526199, chr1:114,679,616 *hg38*, c.860 T; NCBI reference sequence NM_000036.3) is present not only in the three high-coverage Neandertal genomes but also in seven additional Neandertal genomes[9,13–19] sequenced at lower coverage where this genomic position is observed (Supplementary Table 2). In contrast, it is absent from the Denisovan genome[20] and from other primate genomes (Fig. 1b, c), suggesting that it is specific to Neandertals.

This nucleotide substitution results in the replacement of a lysine residue by an isoleucine residue at position 287 in AMPD1 (p.K287I; NCBI reference sequence NP_000027.3, previously also reported as c.A860T and p.K320I[21–24]). In mammals, AMP deaminases are encoded by a multigene family with tissue-specific expression patterns[5,25–28]. *AMPD1* is expressed predominantly in skeletal muscle (Fig. 1d), while the related *AMPD2* and *AMPD3* genes, originally identified as liver- and erythrocyte-specific isoforms, respectively, are expressed at low levels in several tissues (Supplementary Fig. 2a). The AMPD2 and AMPD3 proteins share 51% and 61% amino acid sequence identity with AMPD1, respectively.

### K287I affects a highly conserved position in AMPD1

AMPD1 has distinct regulatory and functional domains[25,29,30] (Fig. 2a). Alignment of AMPD1 homologs from 246 species spanning various animal kingdoms shows that the N-terminal one-third of the protein is less conserved than the C-terminal two-thirds (Fig. 2c), congruent with previous findings[30–32]. Lysine at position 287 in AMPD1 is conserved from yeast to vertebrates as well as in AMPD2 and AMPD3 (Fig. 2d) suggesting that it may have an important functional role. As no crystal structure of human AMPD1 has been described, we modeled the predicted structure of AMPD1[33,34] on the X-ray structure of human AMPD2[35] (Fig. 2b, e). AMP deaminases function as homo-/hetero-tetramers[25,36–38]. Interestingly, the lysine residue at position 287 is inferred to form a salt bridge to the aspartate residue at position 361 (D361) at the interface of the two AMPD1 units forming a dimer (Fig. 2f, g). The D361 residue, like K287, is conserved in eukaryotes as well as in AMPD2 and AMPD3 in humans (Supplementary Fig. 3). A replacement

of the lysine residue for an isoleucine residue would lead to a loss of the salt bridge between the subunits.

### Inferred impact of K287I

All methods for assessing the functional consequences of protein-changing variants currently reported by Ensembl[39] score the K287I as highly impactful (Fig. 3a). To test how its inferred impact compares to other protein-coding changes fixed in Neandertals, we compared one of these impact scores for all 449 derived missense variants that are homozygously present in the three high-coverage Neandertal genomes but absent in the Denisovan genome or among the African genomes in the 1000 Genomes Project[12] (1kGP). The K287I substitution in AMPD1 is predicted to be the most impactful of these substitutions (REVEL score 0.881; Fig. 3b).

### K287I reduces AMPD1 activity in vitro

To investigate the impact of the K287I amino acid substitution on the activity of AMPD1, we expressed the Neandertal (I287) and modern (K287) versions of the protein in human Expi293F cells (Supplementary Fig. 4) and tested the activities of the purified proteins using a previously described assay[40]. In brief, in this assay the conversion of AMP to IMP catalyzed by AMPD1 is measured by the conversion of the latter compound to xanthosine monophosphate (XMP) by IMP dehydrogenase (IMPDH), resulting in the reduction of oxidized nicotine adenine dinucleotide (NAD⁺) to NADH, which is monitored by an increase in the absorbance at 340 nm (Supplementary Fig. 5).

AMP deaminases have been described to exhibit Michaelis-Menten kinetics as well as other regulatory kinetic behaviors[25,29,41–45]. In our analysis, a substrate inhibition model, which accounts for decreased enzyme activity at higher substrate concentrations and is the most common deviation from Michaelis-Menten kinetics[46], fit the data best (Fig. 3c) (see **Methods**). We found that the kinetics between Neandertal AMPD1 and the modern AMPD1 were significantly different ($p = 6.9 \times 10^{-21}$). The apparent maximum enzyme activity of the Neandertal AMPD1 was ~25% lower than the activity of the modern AMPD1 (Neandertal $V_{max}$: 19.21 ± 2.16 μmol min⁻¹ mg⁻¹ protein, 95% CI 14.90–23.51; modern $V_{max}$: 24.96 ± 2.58 μmol min⁻¹ mg⁻¹, 95% CI 19.83–30.09; mean ± SEM). Substrate affinity, as measured by the Michaelis-Menten constant $K_m$, was lower for the Neandertal protein but did not differ significantly between the enzymes ($K_m$ for the Neandertal enzyme: 0.91 ± 0.19 mM, 95% CI 0.53–1.29; $K_m$ for the modern enzyme: 0.61 ± 0.12 mM, 95% CI 0.37–0.85). The values

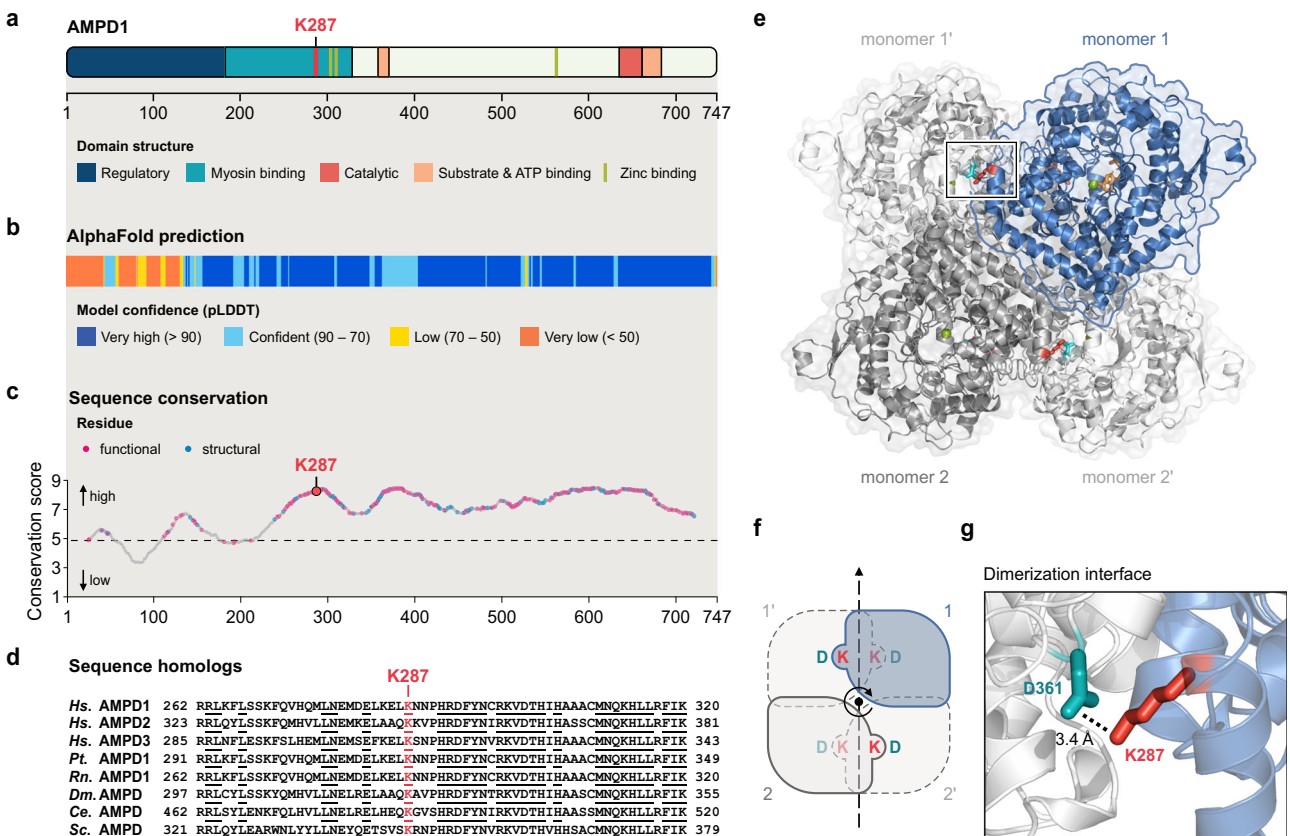

**Fig. 2 | Location of the K287I amino acid replacement in human AMPD1.**
**a** Domain structure of human AMPD1. The scale indicates amino acid positions.
Color coding: K287, red; regulatory domain, dark blue; myosin binding domain,
teal; catalytic domain, light red; substrate and ATP binding domain, light orange;
Zinc binding motif, light green. **b** AlphaFold prediction of AMPD1 corresponding to
the indicated amino acid positions in panel a. The confidence level of the prediction
is color-coded by the per-residue Local Distance Difference Test (pLDDT) con-
fidence score. **c** Sequence conservation of AMPD1. Conservation scores as pre-
dicted from multiple sequence alignments of 246 animal AMPD1 homologs,
ranging from 1 (most variable positions) to 9 (most conserved positions). Residues
predicted to be functional and structural are indicated in pink and blue,
respectively. **d** Sequence alignment of AMPD homologs in a highly conserved
region around K287. Residues completely conserved in many species are under-
lined. Hs., Homo sapiens; Pt., Pan troglodytes; Rn., Rattus norvegicus; Ce., Cae-
norhabditis elegans; Dm., Drosophila melanogaster; Sc., Saccharomyces cerevisiae.
**e** 3D structure of human AMPD1 (amino acids 112 to 747) predicted by AlphaFold
and modeled onto the human AMPD2 tetramer (PDB code 8HU6). Root-mean-
square deviation of the alignment is 0.6 Å. AMP depicted as stick model in orange,
$Zn^{2+}$ depicted as green sphere. **f** Cartoon of isologous tetramer assembly of AMPD1
with 2-fold symmetry axis. **g** Inferred salt bridge between K287 (red) and D361
(turquoise) at the dimerization interface of two AMPD1 units. The length of the salt
bridge is 3.4 Å. Source data are provided as a Source Data file.

obtained for $V_{max}$ and $K_m$ were comparable with previously reported
values for modern human AMPD1[25,29,41–43,47].

## K287I reduces AMPD1 activity in vivo

To investigate if the K287I substitution affects the enzyme also when
expressed from the endogenous gene in skeletal muscle, we generated
a mouse line carrying this substitution. We isolated proteins from the
*extensor digitorum longus* muscle from nine transgenic mice of both
sexes as well as eight of their wild-type littermates and measured AMP
deaminase activity as described above (Supplementary Fig. 6).
Although considerable variations were observed among wild-type
individuals of both sexes, the Neandertalized *Ampd1* showed ~80% less
activity than wild-type *Ampd1* (Neandertal vs. wild-type, respectively:
males, $19 \pm 2$ vs. $87 \pm 15$ nmol $min^{-1}$ $mg^{-1}$ protein, $p = 9.0 \times 10^{-4}$; females,
$20 \pm 1$ vs. $93 \pm 17$ nmol $min^{-1}$ $mg^{-1}$, $p = 1.3 \times 10^{-2}$; mean $\pm$ SEM) (Fig. 3d).
Thus, the Neandertal substitution in AMPD1 reduces the activity both
of the purified human recombinant protein and of the mouse protein
when expressed from the endogenous gene in skeletal muscle.

## The Neandertal AMPD1 variant among present-day humans

We next checked for the presence of the K287I substitution among
2504 genomes from the phase 3 release of the 1kGP[12] (Fig. 4a). We find
that the allele resulting in K287I is absent in Africans, East Asians and

African-Americans while it is present at allele frequencies of 2% to 8% in
Europeans (1kGP and[21–23,48]), 1% to 3% in Native Americans, and 1% to 2%
in South Asians.

To investigate whether this substitution could have originated
from gene flow from Neandertals, we examined whether it sits on
haplotypes containing Neandertal-like alleles that are long enough to
exclude the possibility that they have survived recombination from
the common ancestral population of modern and archaic humans.
The missense mutation is commonly inherited on a haplotype of
440 kb (linkage disequilibrium $r^2 > 0.8$ with rs34526199;
chr1:114,371,932–114,812,153, *hg38*) (Fig. 4b) which has a genetic
length[49] of 0.063 centimorgan. Using a published equation[50] and
previously described parameters[51] we show that such a long
genetic segment could not have survived recombination since the
time of the common ancestor of modern humans and Neandertals
($p = 1.7 \times 10^{-10}$).

To confirm Neandertal gene flow, we also examined sequence
similarity by constructing a phylogenetic tree of DNA sequences cov-
ering the core haplotype carrying K287I ($r^2 = 1.0$ with rs34526199;
chr1:114,544,489–114,681,581; 137 kb). The Neandertal-like haplotypes
clustered in a monophyletic group with the three Neandertal genomes
(branch support = 1.0) (Fig. 4c), supporting their introduction into
modern human populations by gene flow from Neandertals.

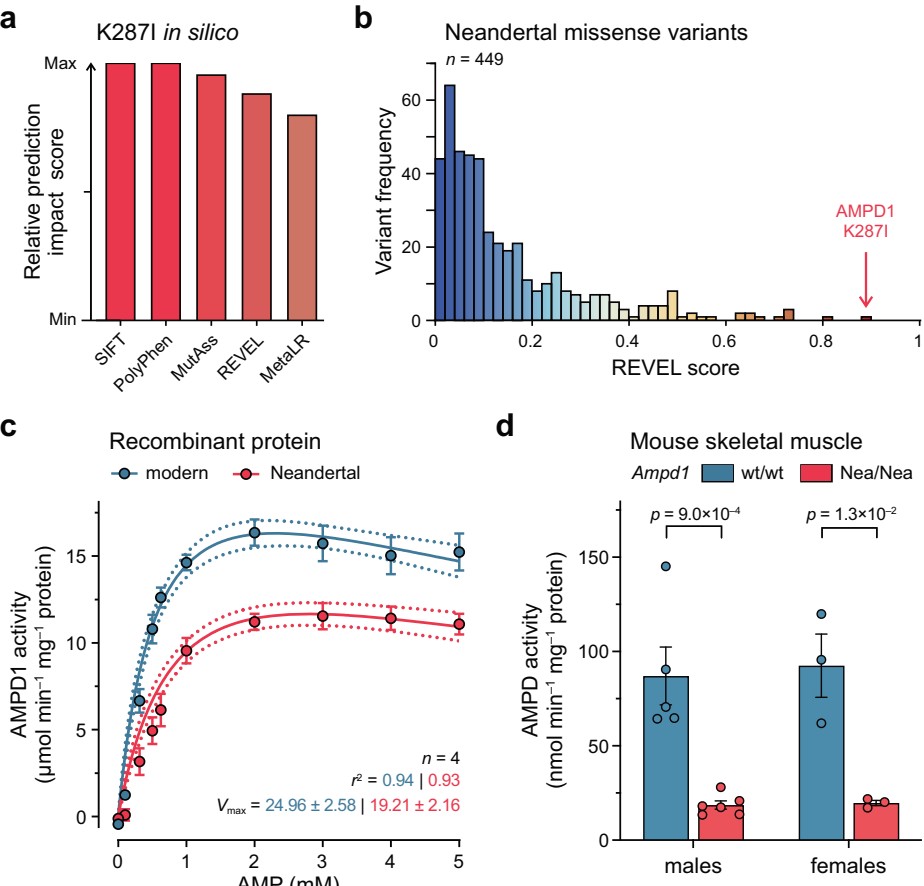

**Fig. 3 | Predicted and measured kinetic effects of the Neandertal K287I substitution in AMPD1. a** In silico predictions of the impact of the K287I substitution. SIFT, Sorting Intolerant From Tolerant; PolyPhen, Polymorphism Phenotyping-2; MutAss, Mutation Assessor; REVEL, Rare Exome Variant Ensemble Learner; MetaLR, Meta Logistic Regression. **b** Distribution of REVEL scores for all ($n = 449$) non-synonymous Neandertal variants. **c** In vitro kinetics of the modern (K287) and the Neandertal (I287) variants of AMPD1. The substrate versus activity curves for recombinant AMPD1 proteins were obtained at 37 °C. $V_{max}$ as µmol min⁻¹ mg⁻¹ protein. Data analyzed using a substrate inhibition model. Dots and error bars represent mean and SEM of $n = 4$ independent measurements. Dotted lines represent 95% confidence intervals. **d** In vivo AMP deaminase activities from protein extracted from *extensor digitorum longus* muscle from transgenic mice carrying the Neandertal (Nea) mutation in *Ampd1* ($n = 6$ males, 3 females) as well as their wild-type (wt) littermates ($n = 5$ males, 3 females). Bars and error bars represent mean and SEM. $p$ values determined using an unpaired two-tailed $t$-test. Source data are provided as a Source Data file.

Some individuals carry longer haplotypes ($r^2 = 0.6$) spanning nine additional genes beyond *AMPD1* (*TRIM33, BCAS2, DENND2C, NRAS, CSDE1, SIKE1, SYCP1, TSHB, TSPAN2*) (Fig. 4b). Among the variants in these genes there are two missense variants in the genes *DENND2C* (rs61752477; $r^2 = 1.0$) and *SYCP1* (rs61730058; $r^2 = 0.74$). However, we note that besides *AMPD1*, only *CSDE1* is expressed to an appreciable extent in skeletal muscle (Supplementary Fig. 2b).

**Phenotypic consequences of AMPD1 deficiency in humans**
Given the high allele frequency of the Neandertal-derived missense variant in Finns (Fig. 4a), we investigated its phenotypic consequences in the FinnGen biobank[52]. Associations were considered significant only if they remained robust after adjustment for the number of phenotypes tested ($n = 2469$). To ensure that the observed associations originated from *AMPD1* rather than other genes on the haplotype (Fig. 4b), we required replication using an independent loss-of-function variant. For this purpose, we used a C-to-T substitution at position 34 in the *AMPD1* coding region (c.C34T, rs17602729), which introduces a stop codon at position 12 of the translated protein (p.Q12X), leading to premature translation termination (Fig. 5a). This substitution occurs at a frequency of 9–14% in Europeans (1kGP and[53–55]) and is the most common cause of AMPD1 deficiency in this population[3,6,56]. It is inherited

independently of the p.K287I variant in the population ($r^2 < 0.01$ in the 1kGP).

Using the criteria described above, we found "diseases of veins, lymphatic vessels, and lymph nodes (not elsewhere classified)" and "other disorders of veins" to be positively associated with the Neandertal *AMPD1* haplotype ($p = 4.7 \times 10^{-7}$ and $p = 1.5 \times 10^{-5}$, respectively) and also with c.C34T ($p = 3.0 \times 10^{-3}$ and $p = 9.2 \times 10^{-3}$, respectively). Both phenotypes are broad categories encompassing multiple specific conditions. Notably, both the Neandertal haplotype and c.C34T were associated with "varicose veins" ($p = 1.2 \times 10^{-4}$ and $p = 6.5 \times 10^{-3}$, respectively), suggesting that varicose veins may be a key contributor to the observed associations. We further investigated these associations in the UK Biobank and replicated the association for c.C34T ($p = 1.3 \times 10^{-2}$) but not for the Neandertal *AMPD1* haplotype ($p = 0.26$). Taken together, carriers of variants that reduce AMPD1 activity have an estimated 3–6% increased risk of varicose veins (Fig. 5b).

Having established that the p.K287I substitution reduces AMPD activity in recombinant proteins and in muscle biopsies from transgenic mice, we next sought to determine if we could find similar evidence in present-day humans. To this end, we reviewed the literature for published case studies. One report described an individual who was compound heterozygous for the p.K287I and c.C34T variants. Skeletal muscle AMPD activity in this individual was approximately 41% of that

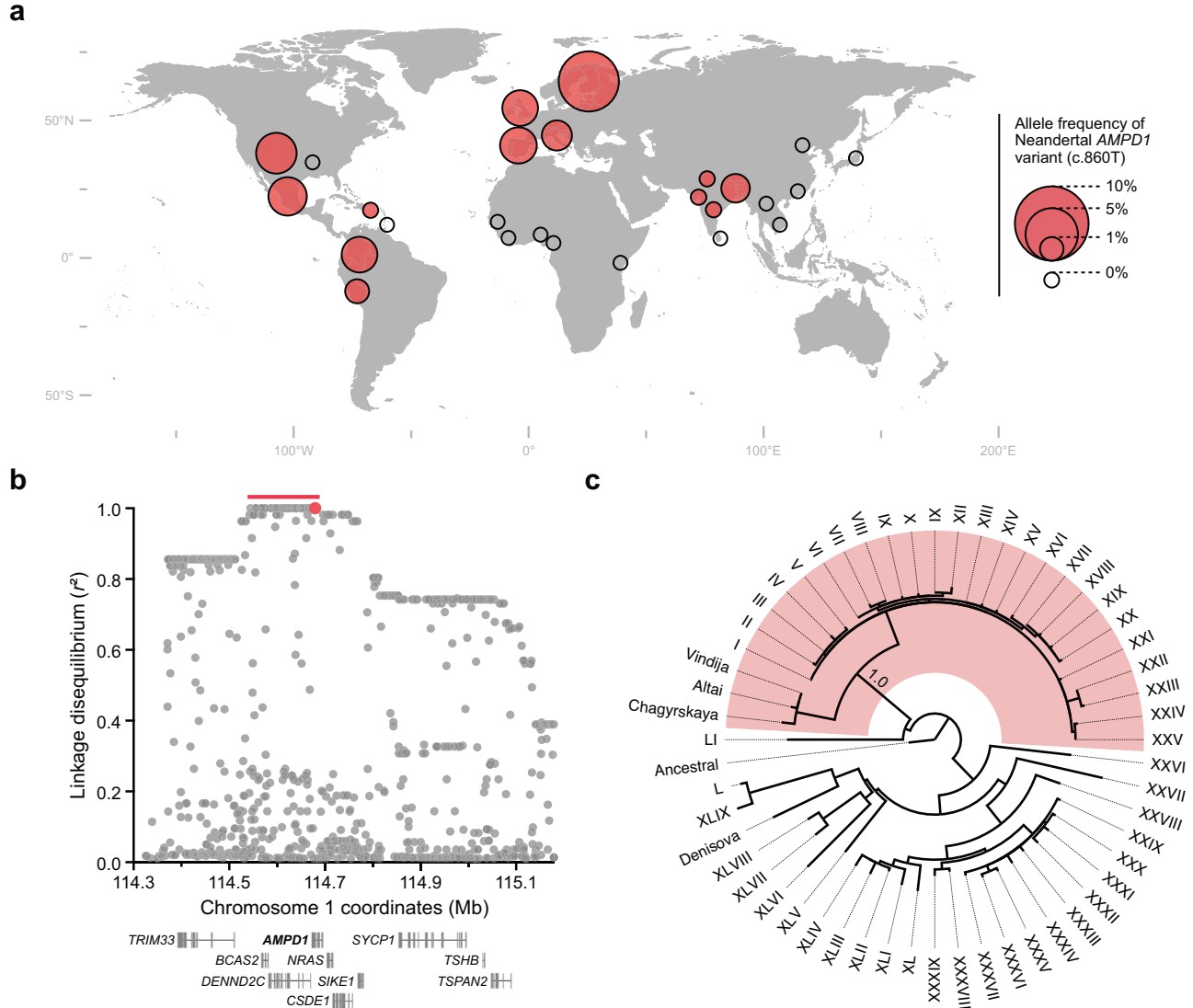

**Fig. 4 | Gene flow from Neandertals introducing the Neandertal AMPD1 into the gene pool of modern humans. a** Geographic distribution of the allele frequency of the Neandertal-like allele in *AMPD1* (c.860T, rs34526199) in 26 populations from 1kGP, displayed by red disks with the scale as shown. Open circles indicate populations without the allele. **b** Co-inheritance between the Neandertal-like allele (c.860T, rs34526199, red dot) and other variants in the surrounding genomic region on chromosome 1. Core haplotype ($r^2 = 1.0$ with rs34526199, 137 kb) indicated by red bar. The x axis shows *hg38* coordinates. Genes in the region are indicated below using standard gene symbols. **c** Phylogeny relating Neandertals and Denisovans with modern human haplotypes (roman numerals). The red shaded area marks all haplotypes carrying the Neandertal missense variant. The phylogeny is rooted with the inferred ancestral sequence. Source data are provided as a Source Data file.

observed in c.C34T heterozygotes[23] (Fig. 5c). This would suggest that the activity of the enzyme carrying the p.K287I substitution would be approximately half of that of the ancestral enzyme. Another compound heterozygote has been reported carrying the p.K287I substitution together with a 4 base pair deletion in intron 2 (rs398123114; absent in the 1kGP populations) which results in alternatively spliced mRNA transcripts. This individual had similar AMPD1 activity as the individual carrying p.K287I and the knockout c.C34T mutation. In the latter case report, the individual suffered from chronic muscle pain, cramps, and exercise intolerance[22].

Due to the low frequency of the Neandertal variant among Europeans, we were unable to directly assess its impact on muscle performance in humans. However, the c.C34T substitution in *AMPD1* has previously been associated with reduced athletic performance. To provide a comprehensive evaluation of whether reduced AMPD1 activity affects athletic outcomes, we performed a meta-analysis (see **Methods**) of seven studies analyzing the prevalence of the c.C34T substitution among 1208 elite athletes and 1537 sedentary controls[57–63]. The frequency of 34T varies from 4% to 14% in athletes and from 9% to 19% in controls. For endurance sports, we found that 34T is significantly negatively associated with being an athlete (Fig. 5d; OR = 0.55, 95% CI: 0.43–0.70, $p = 1.7 \times 10^{-6}$). Similarly, we observed a negative association for power-oriented disciplines (Fig. 5e; OR = 0.49, 95% CI: 0.37–0.65, $p = 7.8 \times 10^{-7}$). We note that loss of AMPD1 function seems to affect the likelihood of being an athlete in both endurance and power sports to a similar extent. Carrying one dysfunctional AMPD1 allele confers approximately a 50% lower probability of achieving elite athletic performance.

### Phenotypic consequences of AMPD1 deficiency in mice

*Ampd1* knockout mouse models have been developed as proxies for human AMPD1 deficiency[64–66]. These studies have not reported significant differences in phenotype, behavior, or exercise performance. To further investigate potential phenotypic and behavioral effects, we

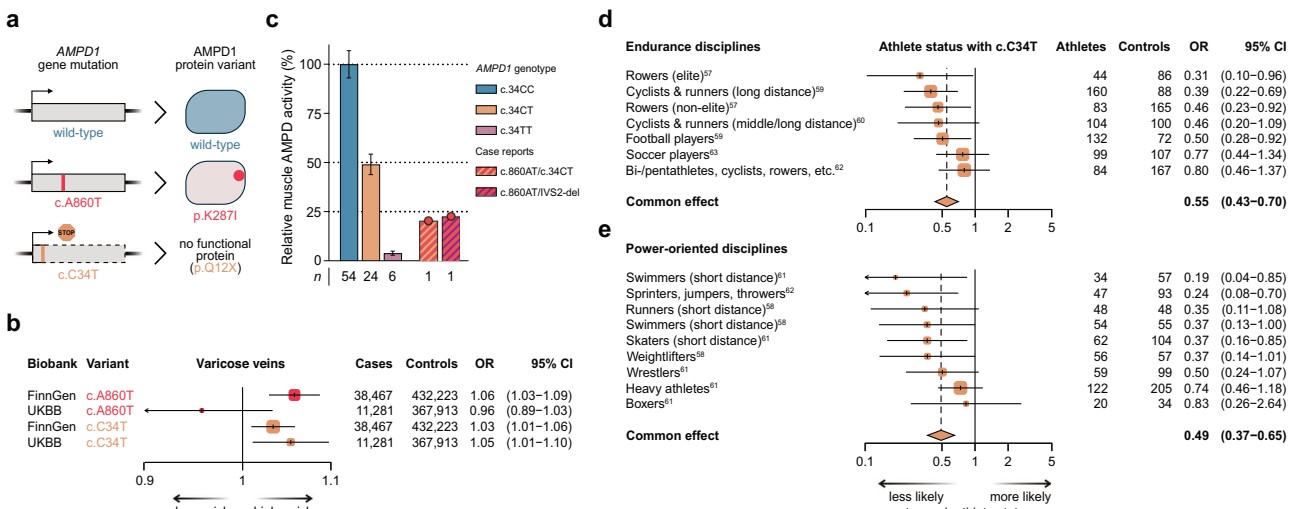

**Fig. 5 | Phenotypic consequences of reduced AMPD1 activity. a** Schematic of *AMPD1* gene mutations and the resulting gene products. The Neandertal missense mutation (c.A860T, rs34526199) results in AMPD1 (p.K287I) with reduced activity, while the c.C34T mutation (rs17602729) leads to a premature stop codon and a non-functional protein (AMPD1 p.Q12X). **b** Both the c.A860T and the c.C34T variants in *AMPD1* are generally associated with an increased risk of varicose veins in individuals in the FinnGen and UK biobanks. Box centers indicate the mean odds ratio (OR), with box sizes proportional to study weight. Horizontal lines represent 95% confidence intervals. **c** In vivo measurements of total skeletal muscle AMP deaminase activity in wild-type controls (c.34CC), carriers of the premature stop codon in *AMPD1* (c.34CT and TT), and two heterozygous carriers of the Neandertal

substitution (c.860AT). One of these individuals was a compound heterozygote for c.860AT and c.34CT, while the other one carried c.860AT and a deletion in intron 2 (IVS2-del, rs398123114). Data from case reports[22,23]. Bars with errors represent mean ± SEM. Forest plots showing odds ratios for attaining athlete status in endurance (**d**; *n* = 706 athletes, 785 controls) and power-oriented sports (**e**; *n* = 502 athletes, 752 controls) among carriers of the common *AMPD1* knockout allele (c.C34T). Athletes from studies indicated by superscript reference numbers[57–63]. Box centers indicate the mean odds ratio, with box sizes proportional to study weight. Horizontal lines represent 95% confidence intervals. Source data are provided as a Source Data file.

analyzed data from the International Mouse Phenotyping Consortium[67] (IMPC) which currently comprises tens of thousands of mice, including 179 heterozygous and 22 homozygous *Ampd1* knockout mice. Mice carrying one copy of an *Ampd1* knockout allele had 10% (2.3 g) higher body weight than wild-type controls ($p = 2.7 \times 10^{-23}$). While this increase remains within the variation observed in controls, the difference is reflected in the growth curves of mutants and wild-types which differ significantly (males $p = 2.0 \times 10^{-48}$, females $p = 6.1 \times 10^{-81}$) (Supplementary Fig. 7a,b). In terms of body composition, heterozygous *Ampd1* knockout mice have a 26% (1.5 g) higher fat mass ($p = 2.0 \times 10^{-4}$), also reflected in a reduced lean-to-body weight ratio ($p = 1.8 \times 10^{-3}$), and an 8% lower bone mineral content relative to body weight ($p = 5.0 \times 10^{-4}$) (Supplementary Fig. 7c and Supplementary Table 3). These differences remain significant after correction for multiple comparisons. No skeletal muscle histopathology is reported. Heterozygous *Ampd1* knockouts show an 8% reduced forelimb grip strength relative to body weight compared to wild-type controls ($p = 1.9 \times 10^{-2}$), but show no significant differences in activity in an open field test (Supplementary Fig. 8 and Supplementary Table 4).

## Discussion

The p.K287I substitution in AMPD1 which was fixed, or occurred at high frequency, among Neandertals (Fig. 1c; Supplementary Table 2) reduces the activity of the purified human enzyme by approximately 25% (Fig. 3c). In agreement with this, in a previous study with rat AMPD1 enzyme purified from bacteria, the K287I mutation led to a 50% decrease in activity, along with a diminished affinity for the AMP substrate[48]. In one present-day carrier of the Neandertal AMPD1 variant[23], who also carries an inactive AMPD1 allele on the other chromosome, the enzymatic activity of the Neandertal enzyme appears to be reduced by approximately 50% (Fig. 5c). Furthermore, when introduced into the mouse protein, the substitution reduces the AMPD activity by ~80% in muscle extracts (Fig. 3d). Taken together, it is clear that the K287I substitution substantially reduces the activity of

the AMPD1 enzyme ranging from 25–50% in a human background and 50–80% in a murine background. Some of the differences in reduction of activity observed in the different studies may depend on the assays or protein constructs used.

The *AMPD1* gene is conserved among vertebrates. In fact, in zebrafish, its expression is restricted to skeletal muscle, as in humans, suggesting that AMPD1 has played an important role in skeletal muscle physiology among vertebrates for ~400 million years or more (Supplementary Fig. 2c). It is therefore surprising that the AMPD1 variant described here has risen to what seems to be fixation or near fixation in Neandertals. This is also striking given that *AMPD2* is not expressed in skeletal muscle and *AMPD3* is expressed at 13% of the mRNA levels of *AMPD1* (Supplementary Fig. 2a), limiting the possibility of these enzymes to compensate for the reduced activity of AMPD1. The Neandertal AMPD1 variant is not higher expressed than the majority variant (Supplementary Fig. 9), indicating that the reduced enzyme activity is not compensated by *cis*-acting elements on the Neandertal haplotype. Likewise, the loss of AMPD1 function is not compensated by the expression of *AMPD2* or *AMPD3* in these carriers, nor by the expression of 20 additional genes involved in AMP metabolism, as annotated in the Gene Ontology[68] (GO:0046033). Similarly, AMPD1-deficient individuals (c.34CT or TT) also showed no differences in the expression of these genes (Supplementary Fig. 10).

The amino acid substitution that reduces AMPD1 activity appears to have risen to high frequency among Neandertals (Supplementary Table 2), suggesting that lowered AMPD1 activity was tolerated in this hominin group. The fact that the Neandertal AMPD1 variant, as well as the inactivating C34T mutation that occurs homozygously in ~1% of Europeans, exist in present-day people, suggests that skeletal muscle function in modern humans remains relatively unaffected by reduced AMPD1 activity. Notably, mutations that reduce AMPD1 activity have occurred twice in western Eurasia, first in Neandertals and then later in modern humans, as manifested by the C34T mutation. In contrast, among 103 sequenced placental mammals, an orthologue of *AMPD1*

has been identified in 90 species[69]. Taken together, these data suggest that purifying selection acting on *AMPD1* has been relaxed in Neandertals and in modern humans. Indeed, in a dataset of 807,162 genomes (gnomAD v4.1, ref. [70]), the number of predicted loss-of-function variants ($n = 71$) in *AMPD1* is not significantly different from what would be expected if the gene evolved like a pseudogene ($n = 84$).

AMPD1 deficiency demonstrates incomplete penetrance; the ones affected present with exercise-induced muscle symptoms such as early fatigue, cramps, and/or myalgia[3,54]. Presently, the clinical significance of AMPD1 deficiency remains unclear[4,71–75]. Some studies[60,72,75,76] have found no effect of AMPD1 deficiency on muscle performance, while other studies have demonstrated associations, such as reduced exercise capacity[73]. In a study using a cycling test, which induces a significant depletion of ATP and accumulation of IMP, healthy AMPD1-deficient subjects exhibited a 10% lower mean power output and earlier fatigue development[77]. In patients with coronary artery disease, carriers of the c.34T allele had lower peak oxygen uptake and a diminished response to endurance training[78]. AMPD1 knockouts (c.34TT) also showed earlier fatigue and reduced capacity for voluntary and electrically stimulated repetitive submaximal isometric muscle contractions[71]. Moreover, the presence of the 34T allele is associated with increased risk of ankle, knee, and total injuries, irrespective of the professional level[79], an observation seen also in elite endurance athletes[80] and professional football players[81]. Adding to the spectrum of clinical manifestations of AMPD1 deficiency, we found that both the knockout mutation (c.C34T) and the enzyme activity-reducing Neandertal mutation (c.A860T) in AMPD1 are significantly associated with an increased risk of varicose veins (Fig. 5b). However, this association did not replicate in the UK Biobank, where a large confidence interval suggests this association test was statistically underpowered. The varicose vein association may reflect a connection between impaired muscle energy metabolism and vascular function[82–84]. The meta-analysis of athletic performance performed here (Fig. 5d, e) suggests that AMPD1 deficiency is associated with a lower probability of becoming a top athlete. In support of this notion, *Ampd1* knockout mice exhibit increased body weight and fat mass, along with relatively lower forelimb grip strength (Supplementary Figs. 7 and 8a). In the general population we find no effect on body composition or any performance-related phenotype, neither for c.C34T nor for c.A860T. However, in a study of 154 Lithuanian top-athletes, the c.C34T variant is associated with reduced muscle power and increased fat mass[62], in accordance with the phenotypes observed in mice. In summary, although AMPD1 activity is of importance under 'extreme' conditions and for top athletes, the enzyme seems to be of only minor or moderate importance for normal human physiology in a contemporary Western society.

The reason for the relaxation of purifying selection of AMPD1 remains unknown. One possibility is that the smaller effective population size of Neandertals[85] reduced the effectiveness of selection, making it easier for genetic variants to become fixed[86]. Another possibility is that alternative pathways emerged to replenish ATP during exercise, reducing the need for AMPD1. Additionally, it is possible that cultural and technological advancements in modern humans, Neandertals, and their common ancestor reduced their reliance on extreme muscle performance.

## Methods

### Ethics

As only publicly available data was used, this study did not require a specific ethical permit. Throughout the project, we followed all relevant ethical guidelines and rules. Regarding human data, only publicly available statistics from previous analyses of the UK Biobank[87] (UKBB) and FinnGen[52] studies were used. The UKBB was approved by the National Health Service North-West Center Research Ethics Committee. Study subjects in FinnGen provided informed consent for biobank research, based on the Finnish Biobank Act. All animal experiments

and recombinant DNA procedures were approved by the Okinawa Institute of Science and Technology Graduate University (OIST) Animal Care and Use Committee (ACUP-2022-017-3) and OIST Biosafety Committee (RDE-2020-012-9).

### Bioinformatics

Protein and DNA sequences of AMPD1 were compared using the University of California, Santa Cruz (UCSC) Genome Browser (http://genome.ucsc.edu/). RNA expression data were accessed from The Human Protein Atlas[88] (https://www.proteinatlas.org) on 13 March 2023, and The Genotype-Tissue Expression project[89] (GTEx V10, https://www.gtexportal.org/home/). Linkage disequilibrium between rs34526199 and polymorphisms of other genes located in its vicinity was analyzed via the National Cancer Institute's LDproxy tool[90] (https://ldlink.nci.nih.gov/), using GRCh38 high coverage genomes of all 1kGP populations[12].

The probability of incomplete lineage sorting was modeled using a mathematical framework[50], with previously described parameters[51] using the genetic length and branch lengths in generations, as follows. We estimated the expected genetic length of a shared ancestral sequence given by the inverse of the total branch length. Assuming ~21,500 generations since the common ancestor of Neandertals and modern humans[91] and archaic admixture ~2000 generations ago[92], the total branch length under a recent gene-flow model is $2 \times 21{,}500 - 2000 = 41{,}000$ generations, yielding $L = 1/410$ cM. Conditioning on observing the archaic like tract on both branches, the probability of a length of at least length $m$ follows a Gamma distribution (shape = 2, rate = $1/L$). Using a genetic length of 0.063 cM calculated from the physical distance ($r^2 = 0.8$ with rs34526199; chr1:114,371,932–114,812,153, *hg38*) and a recombination map from deCODE[49], we solved numerically the equation 1-GammaCDF($m$, shape = 2, rate = 410) giving a significance threshold of $p = 1.7 \times 10^{-10}$.

Phylogenies were estimated on archaic and present-day haplotypes spanning the genomic region defined by the Neandertal core haplotype ($r^2 = 1.0$ with rs34526199) using bi-allelic single-nucleotide variants. The phylogenetic analysis included four high-coverage archaic genomes, all haplotypes carrying the Neandertal-derived allele at rs34526199, and one representative haplotype from each 1kGP population with the ancestral state at rs34526199. Positions with a heterozygous call in any of the archaics were excluded, and missing calls in the archaics were imputed to the modern human reference sequence. The inferred ancestral states at variable positions in modern humans were taken from Ensembl[39]. The phylogeny was then inferred using neighbor-joining as implemented in Muscle v5[93]. A phylogenetic tree was generated using the University of Edinburgh's FigTree software (http://tree.bio.ed.ac.uk/software/figtree/).

Single-cell RNA expression in zebrafish[94] was analyzed, and a UMAP plot was generated using the UCSC Cell Browser[95] (https://cells.ucsc.edu). 3D protein structures were generated using PyMOL (The PyMOL Molecular Graphics System, version 2.3.0, Schrödinger, LLC.).

### Phenotypic consequences

Phenotypic associations were investigated in the UK Biobank[87] (https://pheweb.org/UKB-TOPMed/) and in FinnGen[52] (release 12, https://r12.finngen.fi/).

For the meta-analysis of the premature stop codon variant in *AMPD1* (c.C34T, rs17602729) and athletic performance, sports disciplines were categorized as either "endurance" or "power". To avoid inflating statistics by repeatedly using the same controls, controls were allocated proportionally across sports within each study. To maintain integer counts of controls while preserving approximate allele frequencies, the Sainte-Laguë method was applied to distribute genotypes across sports disciplines. Logistic regression was used for association testing, followed by separate meta-analyses for endurance and power groups using standard inverse-variance weighting with a

fixed common effect. Studies with fewer than 20 athletes or without clear classification as "endurance" or "power" were excluded.

Phenotypic data for *Ampd1* mutant mice were obtained from the International Mouse Phenotyping Consortium[67] (IMPC, https://www.mousephenotype.org) on 18 February 2025. All available phenotypes within the classified procedures "Body Weight", "Body Composition (DEXA lean/fat)", "Grip Strength", and "Open Field" were analyzed, including only datasets where at least three mice per group were tested. All mutants carried the *Ampd1*^*tm1b(KOMP)Wtsi*^ allele in either heterozygous or homozygous form and were classified as being in the early adult life stage.

## Multiple Sequence Alignment
The AMPD1 protein sequence (RefSeq NP_000027.3) was used as a query for sequence alignments via the National Library of Medicine's Protein BLAST tool (https://blast.ncbi.nlm.nih.gov/) and for calculating conservation scores using the ConSurf software[96]. We used the blastp algorithm for searching homologs (250 maximum aligned sequences from nr_clustered protein database) and used 246 sequences that sample the list of homologs to the query (with a minimal 35% and a maximal 95% sequence identity between sequences, E-value cutoff 0.0001). Conservation scores were calculated with the Bayesian method (amino acid substitution model was chosen by best fit). We plotted the cumulative conservation score using a 50-amino-acid sliding window.

## Cloning, expression, and purification of recombinant proteins
Recombinant AMPD1 (modern and K287I) was produced by the Protein Science Facility (PSF) at Karolinska Institutet. The coding sequence for human AMPD1 (amino acids 111-747, Δ110 N-truncated, RefSeq NP_000027.3), with and without the mutation K287I, was cloned into pcDNA3.1 expression plasmids by standard cloning methods. The constructs carried a C-terminal 3C-TwinStrep-TEV-GFP fusion. Expi293F suspension cells (Thermo Fisher Scientific, A14527) were transfected with the expression plasmids using FectoPRO transfection reagent (Polyplus). Three days post-transfection, cells were harvested, and cell pellets were frozen in liquid nitrogen and stored at −80°C. For protein purification, cell pellets were thawed and solubilized in buffer A (50 mM HEPES, 100 mM NaCl, 5 mM MgCl$_2$, 10% glycerol, 1% DDM, pH 8), supplemented with BioLock (IBA LifeSciences) and complete protease inhibitor cocktail (Roche), followed by centrifugation to remove insoluble material. TCEP was added to the supernatant to a final concentration of 0.5 mM. Also, the NaCl concentration was adjusted to 500 mM before the supernatant was loaded onto 0.6 ml Strep-Tactin XT resin (IBA LifeSciences) pre-equilibrated with buffer B (100 mM Tris-Cl, 500 mM NaCl, 1 mM EDTA, 10% glycerol, 0.5 mM TCEP, pH 8). The Strep-Tactin XT resin was washed with buffer B, and bound proteins were eluted in buffer C (100 mM Tris-Cl, 250 mM NaCl, 100 mM glycine, 10% glycerol, 1 mM EDTA, 50 mM biotin, 0.5 mM TCEP, pH 8). Extra TCEP was added to the purified proteins to a final concentration of 2 mM. Purification batches were aliquoted, flash-frozen in liquid nitrogen, and stored at −80 °C.

Protein purity and size (Supplementary Fig. 4a,c-e) were assessed using the Bioanalyzer Protein 230 kit (Agilent Technologies, 5067-1517), and concentrations (Supplementary Fig. 4f) were measured with a Bradford protein assay (Bio-Rad, 500-0006). In short, absorbance at 595 nm was measured for bovine serum albumin (BSA; Carl Roth, 8076) standards and recombinant AMPD1 proteins in a flat-bottom 96-well plate (Corning, 3635) using a microplate reader (CLARIOstar^Plus^, BMG Labtech). Protein concentrations were calculated from the BSA calibration curve.

## Mass spectrometry of recombinant proteins
Intact protein samples were run on a ACQUITY Xevo G2-XS (Waters) liquid chromatography mass spectrometry system. Chromatographic separation was achieved on a BioResolve RP mAb Polyphenyl column

(450 Å, 2.7 μm, 2.1 × 150 mm; Waters, 186008946) using water as mobile phase A and acetonitrile as mobile phase B, both with 0.1% difluoroacetic acid. The gradient was 10% B (0.5 min), increased to 90% B (5.5 min), wash step at 90% B (1 min), and re-equilibration at 10% B (1.4 min), for a total run time of 8.5 min. Flow rate was 0.2 ml/min, with 2 μl injections of each sample using MassLynx v4.2 (Waters). Two blanks were run prior to, and one after, the samples. The Xevo G2-XS QTof MS operated in electrospray mode (capillary voltage 3 kV), with lock mass (reference compound leucine enkephalin) acquired every 45 s per manufacturer's protocol for automatic mass correction. Protein-containing chromatographic peaks were combined to generate multiple charged mass spectra and deconvoluted to neutral masses using MaxEnt I software (Waters). Initial survey deconvolution was performed over a wide mass range at 20 Da resolution, followed by a narrower range at 1 Da resolution, suitable for the proteins. Analysis of the purified proteins identified protein masses that correspond to chromophore maturation (GFP) and acetylated forms of AMPD1 (Supplementary Fig. 4b).

## Generation of engineered *Ampd1* mice
Mice carrying Neandertalized *Ampd1* (p.K285I in NCBI reference sequence NP_001028475) were generated using the CRISPR/Cas9 system on a C57BL/6 N (The Jackson Laboratory Japan) mouse strain by the Laboratory Animal Resource Center in Transborder Medical Research Center, University of Tsukuba. A 90-mer oligonucleotide was used to introduce nucleotide substitutions (underlined) that changed the lysine codon (AAG) to the isoleucine codon (ATA) present in the Neandertal sequence.

5′-TCAGATGCTCAACGAGATGGATGAGCTGAAGGAGCTGATAAA-CAACCCCCACCGGGACTTTTATAACTGCAGGAAGGTAAGTGTGTCAGC-3′.

The founder animals were backcrossed to the C57BL/6 N strain at OIST for at least five generations before the generation of homozygotes. Genotyping was performed by amplifying DNA extracted from tail samples using the following primers:

*Ampd1*-Fw: 5′-GTCCTGGAGGCTTTGAAATAC-3′
*Ampd1*-Rv: 5′-ACAGCATGGCCTTTCAAATC-3′
*Ampd1*-NeaFw: 5′-ATGAGCTGAAGGAGCTGATA-3′
*Ampd1*-WtRv: 5′-TCCCGGTGGGGGTTGTTCTT-3′.

Mice of the same sex were housed in groups of up to five animals per individually ventilated cage under specific pathogen-free conditions. Nine transgenic mice carrying the Neandertal mutation in *Ampd1* (n = 6 males, 3 females) and eight of their wild-type littermates (n = 5 males, 3 females) were used for experiments. The animals had *ad libitum* access to food and water via an automatic watering system, and environmental parameters, including temperature and humidity, were continuously controlled and monitored. The housing facility operated on a 12:12-hour (8:00/20:00) light–dark cycle. Cages were changed weekly, and animals were monitored daily by trained personnel. For tissue collection, the mice were euthanized with an overdose of inhalant anesthesia (isoflurane), in accordance with institutional and ethical guidelines.

## Preparation of muscle extracts
The *extensor digitorum longus* muscle was collected from the hindlimb of each mouse and snap-frozen in liquid nitrogen. Total proteins were extracted using the Minute Total Protein Extraction Kit for Muscles (Invent Biotechnology, SA-06-MS) following the manufacturer's instructions. The extracts were stored at −80°C before subsequent experiments. Protein concentrations were determined by Pierce BCA assay (Thermo Fisher) (Supplementary Fig. 6a).

## Recombinant enzyme kinetic assays
Kinetic activities of recombinant proteins were acquired using the Continuous AMP Deaminase Assay Kit (Creative Biomart, Kit-0875) by

measuring absorbance at a wavelength 340 nm ($A_{340}$) in a microplate reader (CLARIOstar$^{Plus}$, BMG Labtech) with a round-bottom 96-well plate (Corning, 3797). AMPD1 was assayed at 0.4 µg of enzyme per reaction, by adding 5 µl of protein dilution to 200 µl reaction buffer (containing 100 mM Tris-HCl, 100 mM KCl, 12 mM MgCl$_2$, 10 mM DTT, 10 mM NAD, IMPDH > 100 mU/ml, pH 8.1), and the reaction components were equilibrated for 15 min at 37 °C. To start the reaction, 10 µl AMP solution (Sigma-Aldrich, 01930) was added at a final concentration varying from 0 mM to 5 mM, followed by immediate measurement of absorbance over 60 min at 37 °C in plate mode, with recordings every 20 s and with 27 flashes, spiral well scan, and settling time 0.1 s.

### Muscle AMPD activity assays
AMPD activities in muscle extracts were measured using the Continuous PRECICE AMP Deaminase Assay Kit (Novocib, K0709-05-2) following the manufacturer's instructions. In brief, 10 µg of total protein from each muscle extract was mixed with 200 µl of reaction buffer containing cofactors (DTT and NAD) and IMPDH in each well of a round-bottom 96-well plate (Corning, 3797). The reaction plate was gently shaken for 1 min and then equilibrated at 37 °C for 15 min. After adding AMP solution (from kit) to a final concentration of 4 mM, the plate was agitated for an additional 1 min, and $A_{340}$ was measured at 37 °C every minute for 120 min using a microplate reader (VICTOR Nivo Multimode, PerkinElmer).

### Kinetic analysis
AMPD activities were quantified by determining the reaction rate (velocity) as the change in $A_{340}$ over time within the linear kinetic range (0–5 min for recombinant proteins, Supplementary Fig. 5; 40–60 min for mouse muscle extracts, Supplementary Fig. 6). For recombinant proteins, kinetic parameters were determined by fitting a plot of velocity $v$ versus substrate concentration $[S]$ to the substrate inhibition model using GraphPad Prism software:

$$v = \frac{V_{max}[S]}{K_m + [S] \cdot \left(\frac{1+[S]}{K_i}\right)} \tag{1}$$

where $V_{max}$ is the maximum enzyme velocity (in the absence of substrate inhibition), $K_m$ is the Michaelis-Menten constant, and $K_i$ is the dissociation constant for substrate binding in the inhibitory state. The Akaike Information Criterion (AIC)[97] was used to select the best-fitting kinetic model. All combinations of $V_{max}$, $K_m$, and $K_i$ were tested, with the optimal model being the one where $K_i$ was shared between the modern and Neandertal AMPD1 enzymes ($K_i$ 8.68 ± 2.75 mM, 95% CI 3.20 – 14.16; mean ± SEM).

The concentration $c$ of produced NADH was calculated using the measured change in $A_{340}$, the extinction coefficient $\varepsilon$ of NADH ($\varepsilon_{340} = 6220$ M$^{-1}$cm$^{-1}$), and an empirically determined path length $d$ of 0.62 cm (for the above used reaction buffers, volumes, and plates, at 37 °C), using the Beer-Lambert law[98]:

$$c_{NADH} = \frac{\Delta A_{340}}{\varepsilon_{340} \cdot d} \tag{2}$$

### Statistics
Graphs in figures were plotted and error bars were calculated using GraphPad Prism 10 software. The number of replicates is stated in the respective figure legends. No statistical method was used to predetermine sample size. The experiments were not randomized. Samples were prepared unblinded but in parallel. Analysis was performed on the basis of numerical sample names, without the identity of the samples being known during the analysis. The Akaike Information Criterion (AIC) was used to select the best-fitting kinetic model for assays with recombinant enzymes. The

significance of differences in mouse muscle AMPD activities was determined using an unpaired two-tailed $t$-test. The significance of differences in body weight of *Ampd1* mutant mice from the IMPC was calculated using a co-dominant model. Significance of all other mouse phenotypes was tested by two-way ANOVA, accounting for sex and *Ampd1* genotype, and adjusted for multiple comparisons where applicable. Effect sizes of mouse phenotypes were calculated as Cohen's $d$.

### Reporting summary
Further information on research design is available in the Nature Portfolio Reporting Summary linked to this article.

## Data availability
The modern human genomes used are available from the 1,000 Genomes Project [https://ftp.1000genomes.ebi.ac.uk/vol1/ftp/data_collections/1000G_2504_high_coverage/], and the Neandertal [http://cdna.eva.mpg.de/neandertal/] and Denisovan [http://cdna.eva.mpg.de/denisova/] genomes from the Max Planck Institute for Evolutionary Anthropology. The ancestral alleles are available at Ensembl [https://ftp.ensembl.org/pub/release-84/fasta/ancestral_alleles/]. The recombination rate data is available as Data S3 from Halldorsson et al.[49]. Biobank data are accessible from the UK Biobank [https://pheweb.org/UKB-TOPMed/] and FinnGen [https://r12.finngen.fi/] using the variant IDs (rs34526199 and rs17602729). Protein structures can be accessed from AlphaFold (human AMPD1, code P23109) and Protein Data Bank (human AMPD2, PDB code 8HU6; *Arabidopsis thaliana* AMPD, PDB code 2A3L). Tissue RNA expression data in humans is available from The Human Protein Atlas (gene code AMPD1, AMPD2, AMPD3 [https://www.proteinatlas.org/]) and The Genotype-Tissue Expression project (V10 [https://www.gtexportal.org/home/downloads/adult-gtex/qtl]). RNA expression in zebrafish was accessed directly from UCSC Cell Browser (gene code ampd1 [https://zebrafish-dev.cells.ucsc.edu/]). Phenotypes in mice were accessed from the International Mouse Phenotyping Consortium (gene code MGI:88015 [https://www.mousephenotype.org/data/genes/MGI:88015]). Source data are provided with this paper.

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

## Acknowledgements

This work was supported by the NOMIS Foundation and the Max Planck Society. H.Z. is supported by the Knut and Alice Wallenberg Foundation (2023.0141), the Swedish Research Council (2021-03050), the Swedish Brain Foundation (PS2022-0040), Groschinskys Minnesfond, as well as Cornell's, Philip-Sörensen's, Åhlén's, and Hedlund's foundations. S.L. is

supported by the internal funding from OIST and JSPS KAKENHI (JP24K18206). We thank Mats Johansson and Anita Lövgren Sandblom at Karolinska Institutet for mass spectrometry analysis of the proteins. We also thank the Laboratory Animal Resource Center in the Transborder Medical Research Center at the University of Tsukuba for generating the transgenic founder mice. Support from the Animal Resource Section at OIST is acknowledged as well. Finally, we want to acknowledge the participants and investigators of the UK Biobank and FinnGen studies.

## Author contributions

H.Z. conceived the idea; D.M., H.Z. and S.P. designed the study; T.N. designed protein constructs and E.S. and H.A.-K. produced recombinant proteins; D.M. performed experiments with recombinant proteins; S.L. bred the mouse strain, optimized genotyping protocols, and performed muscle tissue experiments using the transgenic mice. D.M., S.L. and H.Z. performed data collection and analysis; H.Z. studied the phenotypic effects in biobanks; D.M., H.Z., and S.P. wrote the manuscript with input from all authors.

## Funding

## Competing interests

The authors declare no competing interests.
