## [Transparent Peer Review file · Nature Communications]

Muscle AMP deaminase activity was lower in Neandertals than in modern humans

Corresponding Author: Dr Hugo Zeberg

Version 0:

Reviewer comments:

Reviewer #1

(Remarks to the Author)

In this paper, the authors examined the functional consequences of a Neanderthal-specific mutation in AMPD1 by in vitro, in vivo, and epidemiological analyses. The authors found that this mutant (K287I) in extant humans was introduced from Neanderthals by gene flow. Biochemical analysis showed that this mutant significantly reduced the catalytic activity of AMPD1. AMPD1 activity in muscle extracts was reduced in knock-in mice carrying the Neanderthal mutant. This mutant and another mutant (C34T) in the present human were associated with reduced relative AMPD1 expression and muscle-related phenotypes. Taken together, these findings suggest that AMPD1 is less functionally constrained in Neanderthals and extant humans.

It is very interesting that a nonsynonymous mutant of a gene related to muscle strength has an ancestral human origin. However, the concern underlying this study is that the functional consequences of this mutant have already been reported. Toyama et al. found that introducing K287I into rat AMPD1 significantly reduced enzyme activity. Since the amino acid sequence of AMPD1 is highly conserved among species, the results of this previous experiment may be applicable to the biochemical properties of human AMPD1.

I agree that the present study by Macak provides a fairly detailed analysis of the effects of the mutant. However, the basic findings regarding the biochemical consequences of mutations are not new. Phenotypic analysis of knock-in mice is important to meet the Journal's acceptance criteria, but the authors have only identified the enzymatic activity of AMPD1 in muscle extracts. The authors must present data for further phenotypic analysis, including behavioral examination, histological examination of muscle, and growth curves of knock-in mice. These data are very important in discussing the tolerance of this variant in humans, since the clinical consequences of this variant in modern humans are still unknown and it is difficult to isolate the effects of other variants.

The authors argue that AMPD1 purifying selection is mitigated in Neanderthals and extant humans. However, it has been suggested that natural selection did not work in archaic humans due to their small population size, and that genetic drift may have led to the accumulation of weak mutations.

The last part of the results is difficult to understand because the data are on the AMPD1 knockout allele (C34T), not K283I. The authors should clearly distinguish this data from other data, or else they should create an animal model with that mutation, or a combined heterozygous/homozygous mouse model, and compare phenotypic results.

Reviewer #2

(Remarks to the Author)

"Muscle AMP deaminase activity was lower in Neandertals than in modern humans"

Macak et al. tried to report the characteristics of AMPD1 (adenosine monophosphate deaminase 1) gene variant (lysine-to-isoleucine substitution at position 287; K287I) found in all Neanderthal genomes. This variant, which occurs at an allele frequency of 0-8% outside Africa, was introduced to modern humans by gene flow from Neanderthals. The catalytic activity of the purified Neanderthal AMPD1 is lower than the ancestral enzyme, and when introduced in mice, it reduces AMPD activity in muscle extracts by 80%. Among present-day Europeans, another AMPD1 variant encoding a stop codon occurs at an allele frequency of 9-14%. Individuals heterozygous for this variant are less likely to be top-performing athletes in various

sports, but otherwise reduced AMPD1 activity is well tolerated in present-day humans. Based on these results, while being conserved among vertebrates, they have concluded that AMPD1 seems to have become less functionally important among Neandertals and modern humans.

Comment

Generally, they showed functional changes in this Neandertals K287I variant, and tried to show the difference between Neandertals and modern humans.

They studied the characteristics of the K287I variant by expression of N-terminal deleted enzymes compared with wild type. They showed similar results (reduced activity) as in the previous report in the prokaryotic expression (this report #1 should be cited as well). However, the kinetic properties seem to be different compared with the previous report. They should comment that this might be due to the different expression system (eukaryotic vs prokaryotic, truncated protein vs full length protein)

They also clearly showed gene flow of the K287I variant in modern humans.

Next, they tried to show that the reduced enzyme activities could be related some diseased conditions including varicose veins. They showed significant correlation for the other null variant (Q12X) in two populations, but did show correlation for K287I only in one population. This might be the different characteristics between Q12X and K287I. This should be discussed as well.

They also discussed about the muscle function and these variants, especially in top athlete. Functional change due to these AMPD1 variant might cause some effect but it was not clear since some individual with these variants did not show any muscle symptoms. Therefore, they concluded that AMPD1 seems to have become less functionally important among Neandertals and modern humans. They should cite the previous reports (#2) in the experiment of AMPD1 gene knock-out mice.

#1 Toyama K et al. Haplotype analysis of human AMPD1 gene: origin of common mutant allele. *J Med Genet.* 2004 Jun;41(6):e74. doi: 10.1136/jmg.2003.013151.

PMID: 15173240

#2 Cheng K et al. Effect of isolated AMP deaminase deficiency on skeletal muscle function. *Mol Genet Metab Rep.* 2014 Jan 16;1:51-59. doi: 10.1016/j.ymgmr.2013.12.004. PMID: 27896074

Reviewer #3

(Remarks to the Author)

Macak et al. investigated the evolutionary and functional consequences of the Neandertal-specific AMPD1 variant (K287I). The manuscript is well-structured and addresses an interesting and relevant question in evolutionary genetics. Further, the integrative approach incorporating biochemical assays, transgenic mouse models, population genetics, and phenotypic association analyses provides an impressive set of results to assess the impact of this variant on muscle metabolism and its potential implications for Neandertal physiology. However, the fact that this gene has likely been pseudogenized in the human lineage dampens the impact somewhat. Further, there are several critical areas that require further attention. The authors missed the opportunity to investigate potential epistatic interactions, compensatory mechanisms and RNAseq analysis using transgenic mice, which would have provided a more comprehensive understanding of the functional consequences of the K287I variant. Below, I outline the major and minor points that should be addressed.

MAJOR COMMENTS:

Insufficient reporting of low-coverage ancient genomes:

The authors claim that the K287I variant was fixed in Neandertals based on three high-coverage Neandertal genomes. Their analyses of the low-coverage ancient genomes are not reported in the main text. The claim that the K287I variant is fixed in Neandertals, would be stronger if these low-coverage genome results were incorporated into the main text rather than being left out. Additionally, an analysis of allele frequency changes in ancient genomes over time would be a valuable addition to understanding the evolutionary trajectory of K287I. Within this context, a quantitative assessment about the likelihood of fixation would have been welcome.

Functional interpretation needs more context:

The authors argue that Neandertals had reduced endurance capacity and strength due to lower AMPD1 activity. However, Neandertals are thought to be highly active hunters, and this claim is not fully supported by the data. Thus, the functional consequences of K287I should be interpreted within a broader metabolic and genetic context. Alternative metabolic pathways that could have compensated for lower AMPD1 activity (e.g. mitochondrial adaptations, shifts in energy utilization) are not considered. Epistatic interactions between AMPD1 and other genes involved in muscle function are also ignored. The authors should consider expanding their discussion to include alternative compensatory mechanisms in Neandertals. This includes potential interactions with mitochondrial genes, alternative purine metabolism pathways, and dietary adaptations.

Lack of RNAseq data for transgenic mice:

Related to above point, the study presents evidence for the functional consequences of the Neandertal AMPD1 variant using biochemical assays and transgenic mouse models. However, RNAseq data from transgenic mice would provide crucial insight into gene expression changes, potential compensatory mechanisms, and broader metabolic effects. Without transcriptomic analysis, it is difficult to assess whether other metabolic pathways may compensate for the reduced AMPD1 activity in the transgenic mice. Additionally, RNAseq could help identify secondary effects beyond the immediate enzymatic

deficiency, offering a more comprehensive understanding of the physiological impact of K287I. I regret suggesting additional experiments, but this seems like a very obvious experiment to consider for the high impact claims stated in the manuscript.

Paralog considerations:

The authors mention the two paralogous genes, AMPD2 and AMPD3, without providing any background on these genes or expanding on the functional properties of the paralogs or their sequence similarity. Is it possible that there is functional compensation of AMPD1 loss of function through the increased activity of these paralogs? Moreover, the authors do not account for the possibility that AMPD1 reads may have been mismatched to AMPD2 or AMPD3, assuming they share sequence similarity.

Loss-of-function mutation:

The authors should clearly indicate whether c.C34T carriers also carry the K287I variant and, if so, at what frequency. This would clarify any potential linkage between the two variants. Additionally, Figure 4C should be further annotated to include the distribution of c.C34T carriers.

MINOR COMMENTS:

The manuscript states that 8 primate genomes were used for ancestral state inference, but later Figure 1C refers to 13 genomes. This inconsistency needs to be clarified.

The authors report that the K287I variant is significantly associated with varicose veins in FinnGen, but this association is absent in UK Biobank. The lack of replication is not that surprising due to different approaches, population-specific effects or statistical power issues. However, this issue should be discussed and the authors should weigh in what are the most likely explanations.

Reviewer #4

(Remarks to the Author)

Version 1:

Reviewer comments:

Reviewer #1

(Remarks to the Author)

In the revised version of the manuscript, the authors presented additional data on the phenotypic analysis of Ampd1 knockout mice, which provide valuable insights for comparing the phenotypic consequences of Ampd1 deficiency between humans and mice. I would like to encourage the authors to perform similar phenotypic analyses of Ampd1^{nea/nea} mice in future studies. At present, I believe the authors have adequately addressed the comments, and the revised manuscript meets the criteria for publication.

Reviewer #2

(Remarks to the Author)

"Muscle AMP deaminase activity was lower in Neandertals than in modern humans"

Macak et al. tried to report the characteristics of AMPD1 (adenosine monophosphate deaminase 1) gene variant (lysine-to-isoleucine substitution at position 287; K287I) found in all Neandertal genomes. Individuals heterozygous for this variant are less likely to be top-performing athletes in various sports, but otherwise reduced AMPD1 activity is well tolerated in present-day humans. Based on these results, while being conserved among vertebrates, they have concluded that AMPD1 seems to have become less functionally important among Neandertals and modern humans.

Comment

Revised manuscript was very much improved based on comments by reviewers. Additional data became strengthened the results and their conclusion.

Reviewer #3

(Remarks to the Author)

We appreciate the multiple lines of new analyses and thoughtful responses to our comments. We have no more major comments.

Reviewer #4

(Remarks to the Author)

Muscle AMP deaminase activity was lower in Neandertals than in modern humans

Dominik Macak, Shin-Yu Lee, Tomas Nyman, Henry Ampah-Korsah, Emilia Strandback, Svante Pääbo, Hugo Zeberg

Point-By-Point Response to the Reviewers' Comments

REVIEWER COMMENTS

Reviewer #1 (Remarks to the Author):

In this paper, the authors examined the functional consequences of a Neanderthal-specific mutation in AMPD1 by in vitro, in vivo, and epidemiological analyses. The authors found that this mutant (K287I) in extant humans was introduced from Neanderthals by gene flow. Biochemical analysis showed that this mutant significantly reduced the catalytic activity of AMPD1. AMPD1 activity in muscle extracts was reduced in knock-in mice carrying the Neanderthal mutant. This mutant and another mutant (C34T) in the present human were associated with reduced relative AMPD1 expression and muscle-related phenotypes. Taken together, these findings suggest that AMPD1 is less functionally constrained in Neanderthals and extant humans.

It is very interesting that a nonsynonymous mutant of a gene related to muscle strength has an ancestral human origin. However, the concern underlying this study is that the functional consequences of this mutant have already been reported. Toyama et al. found that introducing K287I into rat AMPD1 significantly reduced enzyme activity. Since the amino acid sequence of AMPD1 is highly conserved among species, the results of this previous experiment may be applicable to the biochemical properties of human AMPD1.

We thank Reviewer #1 for thoughtful comments.

We fully agree with the reviewer that the reduced activity observed for the mutated rat protein suggested a similar effect on the human protein. However, this remained an educated guess that we set out to test experimentally! We much appreciate the study by Toyama et al. and consider it a strength that we could demonstrate the same effect in the human genetic background.

I agree that the present study by Macak provides a fairly detailed analysis of the effects of the mutant. However, the basic findings regarding the biochemical consequences of mutations are not new. Phenotypic analysis of knock-in mice is important to meet the Journal's acceptance criteria, but the authors have only identified the enzymatic activity of AMPD1 in muscle extracts. The authors must present data for further phenotypic analysis, including behavioral examination, histological examination of muscle, and growth curves of knock-in mice. These data are very important in discussing the tolerance of this variant in humans, since the clinical consequences of this variant in modern humans are still unknown and it is difficult to isolate the effects of other variants.

We have now analysed a dataset of 26,244 mice, including 179 heterozygous and 22 homozygous Ampd1 knockout mice. This dataset is part of the International Mouse Phenotyping Consortium (IMPC). As suggested by reviewer, we analysed these data in terms

of physical phenotype, growth curves, and gross histopathology. This is now summarised in a new Supplementary Figure S7 and Supplementary Table 3. For your convenience, we present them here:

Supplementary Fig. S7 | Phenotypic effects of *Ampd1* knockout mice on body weight and body composition. **a** Body weight of heterozygous (het.; $n = 88$ males, 91 females) and homozygous (hom.; $n = 11$ males, 11 females) *Ampd1* mutants compared to wild-type controls (wt; $n = 12,910$ males, 13,133 females). Lines and error bars represent mean \pm SD. **b** Growth curves of body weight of heterozygous *Ampd1* mutants ($n = 2$ to 28 males, 7 to 23 females) compared to wild-type controls ($n = 865$ to 1,802 males, 701 to 1,959 females) between 6 and 17 weeks of age. Data are fitted to a logistic growth curve, with error bars indicating SD and outer lines 95% CI of fit. **c** Body composition phenotypic assays of heterozygous *Ampd1* mutants ($n = 8$ males, 8 females) compared to wild-type controls ($n = 89$ males, 88 females). All mutants were for the *Ampd1^{tm1b(KOMP)Wtsi}* allele and all mice were in the early adult life stage. Data from the International Mouse Phenotyping Consortium (IMPC)⁶⁷. Lines and error bars represent mean \pm SD. p values were calculated by two-way ANOVA, accounting for sex and *Ampd1* genotype, and adjusted for multiple comparisons (p_{adj}) where applicable.

The results show that mice carrying one copy of an *Ampd1* knockout allele had 10% (2.3 g) higher body weight than wild-type controls ($p = 2.7 \times 10^{-23}$). While this increase remains within the variation observed in controls, the difference was also reflected in distinct growth curves between mutants and wild-types (males $p = 2.0 \times 10^{-48}$, females $p = 6.1 \times 10^{-81}$). In terms of body composition, heterozygous *Ampd1* knockout mice have a 26% (1.5 g) higher fat mass ($p = 2.0 \times 10^{-4}$), also reflected in a reduced lean-to-body weight ratio ($p = 1.8 \times 10^{-3}$), and an 8% lower bone mineral content relative to body weight ($p = 5.0 \times 10^{-4}$). These differences remain

significant after correction for multiple comparisons. No significant abnormalities in gross skeletal muscle histopathology were reported by the IMPC.

In terms of behavioural phenotypes, we have analysed grip strength and data from open field tests. These data are summarised in a new Supplementary Figure S8 and Supplementary Table 4. We present it here for your convenience:

Supplementary Fig. S8 | Phenotypic effects of *Ampd1* knockout mice on grip strength and open field test. **a** Grip strength (normalized to body weight) of heterozygous *Ampd1* mutants (het.; $n = 8$ males, 8 females) compared to wild-type controls (wt; $n = 101$ males, 100 females). **b** Open field phenotypic assay of heterozygous *Ampd1* mutants ($n = 8$ males, 8 females) compared to wild-type controls ($n = 99$ males, 100 females). All mutants were for the *Ampd1*^{tm1b(KOMP)Wtsi} allele and all mice were in the early adult life stage. Data from the International Mouse Phenotyping Consortium (IMPC)⁶⁷. Lines and error bars represent mean \pm SD. p values were calculated by two-way ANOVA, accounting for sex and *Ampd1* genotype, and adjusted for multiple comparisons (p_{adj}) where applicable.

Heterozygous Ampd1 knockouts show an 8% reduced forelimb grip strength relative to body weight compared to wild-type controls that is significant after multiple test correction ($p_{adj} = 3.8 \times 10^{-2}$). However, this effect might be conferred by the higher body weight of the mutants, as there were no differences in forelimb grip strength itself. In the open field tests, Ampd1 knockouts show no significant differences.

While we agree that it is valuable to incorporate these data and we now discuss the findings in the manuscript, we also note that mouse models have inherent limitations in capturing human physiology, as has been demonstrated in numerous cases (see amongst others Alquier and Poitout, 2019, PMID: 29143855; von Scheidt et al., 2016, PMID: 27916529).

The authors argue that AMPD1 purifying selection is mitigated in Neanderthals and extant humans. However, it has been suggested that natural selection did not work in archaic humans due to their small population size, and that genetic drift may have led to the accumulation of weak mutations.

We agree with the reviewer that the small population size of Neanderthals would make it more likely that slightly deleterious variants would become fixed. We now address this possibility in the discussion. In other papers, we often mention this scenario, but had missed it here. We thank the reviewer for reminding us!

The last part of the results is difficult to understand because the data are on the AMPD1 knockout allele (C34T), not K283I. The authors should clearly distinguish this data from other data, or else they should create an animal model with that mutation, or a combined heterozygous/homozygous mouse model, and compare phenotypic results.

We have now made an effort to clarify this. If the Neanderthal variant had been more frequent in Europeans (and thus present in the UK Biobank), we would have investigated K287I with respect to athletic ability. In the Discussion, we now discuss the limitations of using the C34T mutation as a proxy for reduced AMPD1 activity of the K287I variant.

Reviewer #2 (Remarks to the Author):

"Muscle AMP deaminase activity was lower in Neanderthals than in modern humans"

Macak et al. tried to report the characteristics of AMPD1 (adenosine monophosphate deaminase 1) gene variant (lysine-to-isoleucine substitution at position 287; K287I) found in all Neanderthal genomes. This variant, which occurs at an allele frequency of 0-8% outside Africa, was introduced to modern humans by gene flow from Neanderthals. The catalytic activity of the purified Neanderthal AMPD1 is lower than the ancestral enzyme, and when introduced in mice, it reduces AMPD activity in muscle extracts by 80%. Among present-day Europeans, another AMPD1 variant encoding a stop codon occurs at an allele frequency of 9-14%. Individuals heterozygous for this variant are less likely to be top-performing athletes in various sports, but otherwise reduced AMPD1 activity is well tolerated in present-day humans. Based on these results, while being conserved among vertebrates, they have concluded that AMPD1 seems to have become less functionally important among Neanderthals and modern humans.

Comment

Generally, they showed functional changes in this Neandertals K287I variant, and tried to show the difference between Neandertals and modern humans.

They studied the characteristics of the K287I variant by expression of N-terminal deleted enzymes compared with wild type. They showed similar results (reduced activity) as in the previous report in the prokaryotic expression (this report #1 should be cited as well). However, the kinetic properties seem to be different compared with the previous report. They should comment that this might be due to the different expression system (eukaryotic vs prokaryotic, truncated protein vs full length protein)

They also clearly showed gene flow of the K287I variant in modern humans.

Next, they tried to show that the reduced enzyme activities could be related some diseased conditions including varicose veins. They showed significant correlation for the other null variant (Q12X) in two populations, but did show correlation for K287I only in one population. This might be the different characteristics between Q12X and K287I. This should be discussed as well.

They also discussed about the muscle function and these variants, especially in top athlete. Functional change due to these AMPD1 variant might cause some effect but it was not clear since some individual with these variants did not show any muscle symptoms. Therefore, they concluded that AMPD1 seems to have become less functionally important among Neandertals and modern humans. They should cite the previous reports (#2) in the experiment of AMPD1 gene knock-out mice.

#1 Toyama K et al. Haplotype analysis of human AMPD1 gene: origin of common mutant allele. J Med Genet. 2004 Jun;41(6):e74. doi: 10.1136/jmg.2003.013151.

PMID: 15173240

#2 Cheng K et al. Effect of isolated AMP deaminase deficiency on skeletal muscle function. Mol Genet Metab Rep. 2014 Jan 16;1:51-59. doi: 10.1016/j.ymgmr.2013.12.004. PMID: 27896074

We thank the reviewer for input and constructive suggestions.

In response to the feedback, we have expanded our discussion on the differences in kinetic properties, emphasizing findings from Toyama et al. Furthermore, we have incorporated new data and a discussion on the phenotypes of Ampd1 knockout mice and cite the work by Cheng et al.

Reviewer #3 (Remarks to the Author):

Macak et al. investigated the evolutionary and functional consequences of the Neandertal-specific AMPD1 variant (K287I). The manuscript is well-structured and addresses an interesting and relevant question in evolutionary genetics. Further, the integrative approach incorporating biochemical assays, transgenic mouse models, population genetics, and phenotypic association analyses provides an impressive set of results to assess the impact of this variant on muscle metabolism and its potential implications for Neandertal physiology. However, the fact that this gene has likely been pseudogenized in the human lineage dampens the impact somewhat. Further, there are several critical areas that require further attention. The authors missed the opportunity to investigate potential epistatic interactions, compensatory mechanisms and RNAseq analysis using transgenic mice, which would have

provided a more comprehensive understanding of the functional consequences of the K287I variant. Below, I outline the major and minor points that should be addressed.

We agree that the functional and molecular consequences of reduced AMPD1 activity on human physiology require further investigation. We have incorporated supporting data as outlined in our responses below.

MAJOR COMMENTS:

Insufficient reporting of low-coverage ancient genomes:

The authors claim that the K287I variant was fixed in Neandertals based on three high-coverage Neandertal genomes. Their analyses of the low-coverage ancient genomes are not reported in the main text. The claim that the K287I variant is fixed in Neandertals, would be stronger if these low-coverage genome results were incorporated into the main text rather than being left out. Additionally, an analysis of allele frequency changes in ancient genomes over time would be a valuable addition to understanding the evolutionary trajectory of K287I. Within this context, a quantitative assessment about the likelihood of fixation would have been welcome.

In addition to the three high-coverage Neandertal genomes, we provide genotype calls for the causative AMPD1 variant in seven additional low-coverage Neandertal genomes, as detailed in Supplementary Table 2. This strengthens the notion that this variant was fixed among Neandertals. We thank the reviewer for this suggestion!

Functional interpretation needs more context:

The authors argue that Neandertals had reduced endurance capacity and strength due to lower AMPD1 activity. However, Neandertals are thought to be highly active hunters, and this claim is not fully supported by the data. Thus, the functional consequences of K287I should be interpreted within a broader metabolic and genetic context. Alternative metabolic pathways that could have compensated for lower AMPD1 activity (e.g. mitochondrial adaptations, shifts in energy utilization) are not considered. Epistatic interactions between AMPD1 and other genes involved in muscle function are also ignored. The authors should consider expanding their discussion to include alternative compensatory mechanisms in Neandertals. This includes potential interactions with mitochondrial genes, alternative purine metabolism pathways, and dietary adaptations.

Following the reviewer's suggestion, we have now extended the discussion to include compensatory mechanisms. To address these in terms of gene expression, we have analysed human transcriptomic data (see below). However, the tolerability of the c.C34T mutation in modern humans suggests that hominins are not as dependent on AMPD1 as other mammals. Furthermore, we prefer to be cautious in interpreting the effects of the p.K287I variant in Neandertals, as the genetic background of Neandertals may influence its effects (as also pointed out by the reviewer).

Lack of RNAseq data for transgenic mice:

Related to above point, the study presents evidence for the functional consequences of the Neandertal AMPD1 variant using biochemical assays and transgenic mouse models. However, RNAseq data from transgenic mice would provide crucial insight into gene expression changes, potential compensatory mechanisms, and broader metabolic effects. Without transcriptomic analysis, it is difficult to assess whether other metabolic pathways may

compensate for the reduced AMPD1 activity in the transgenic mice. Additionally, RNAseq could help identify secondary effects beyond the immediate enzymatic deficiency, offering a more comprehensive understanding of the physiological impact of K287I. I regret suggesting additional experiments, but this seems like a very obvious experiment to consider for the high impact claims stated in the manuscript.

We agree that transcriptomic analysis could provide insights into potential compensatory mechanisms. However, we believe that human data may offer greater relevance, as mouse models have limitations in capturing human phenotypes. To address this, we analyzed transcriptomic data from 816 human skeletal muscle biopsies from the Genotype-Tissue Expression (GTEx) project, including 38 heterozygous carriers of the Neandertal allele and 142 heterozygous and 10 homozygous carriers of the common knockout allele. We focused on the expression of 23 genes involved in AMP metabolism, as annotated in Gene Ontology. We show the data in two new supplementary figures (S9 and S10), and here for your convenience:

Supplementary Fig. S9 | Expression of genes in AMP metabolism in carriers of the Neandertal allele in *AMPD1*. mRNA expression levels of 23 genes involved in the Gene Ontology category “AMP metabolic process” (GO:0046033) in human skeletal muscle biopsies (GTEx V10, ref. ⁹⁶). Data include 778 wild-type (AA) controls and 38 heterozygous (AT) carriers of the Neandertal allele in *AMPD1* (c.860, minor allele frequency: 2.3%). Lines and error bars represent mean \pm SD. p values were obtained from expression quantitative trait locus (eQTL) analysis and adjusted (p_{adj}) for multiple comparisons where applicable.

Supplementary Fig. S10 | Expression of genes in AMP metabolism in carriers of the common knockout allele in *AMPD1*. mRNA expression levels of 23 genes involved in the Gene Ontology category “AMP metabolic process” (GO:0046033) in human skeletal muscle biopsies (GTEx V10, ref. ⁹⁶). Data include 664 wild-type (CC) controls, and 142 heterozygous (CT) and ten homozygous (TT) carriers of the common knockout allele in *AMPD1* (c.34, minor allele frequency: 9.9%). Lines and error bars represent mean \pm SD. *p* values were obtained from expression quantitative trait locus (eQTL) analysis and adjusted (p_{adj}) for multiple comparisons where applicable.

The loss of AMPD1 function is not compensated by the expression of AMPD2 or AMPD3 carriers of the Neandertal allele, nor by changes in expression of 20 additional genes involved in AMP metabolism. Similarly, AMPD1 knockout individuals (c.34CT or TT) showed no differences in the expression of these genes.

In spite of these findings, we now discuss the possibility of compensatory effects at other levels. To fully investigate such potential mechanisms would require a larger study, where a mouse model could be a valuable tool. However, this is beyond the scope of this study.

Paralog considerations:

The authors mention the two paralogous genes, AMPD2 and AMPD3, without providing any background on these genes or expanding on the functional properties of the paralogs or their sequence similarity. Is it possible that there is functional compensation of AMPD1 loss of function through the increased activity of these paralogs? Moreover, the authors do not account for the possibility that AMPD1 reads may have been mismapped to AMPD2 or AMPD3, assuming they share sequence similarity.

We now provide a more detailed description of AMPD2 and AMPD3, including their sequence similarity. Our analysis of human skeletal muscle biopsies (GTEx) shows that AMPD1 loss-of-function is not compensated by increased expression of AMPD2 or AMPD3. Similarly, a previous study found no differences in paralog expression in muscles from Ampd1 knockout versus wild-type mice, as assessed by immunoblotting (Plaideau et al., 2014, PMID: 25459662). While compensatory effects could occur through post-translational modifications or altered enzyme activity, we note that AMPD2 is not expressed in skeletal muscle, and AMPD3 is expressed at only 13% of AMPD1 mRNA levels. We therefore believe that it is unlikely that these paralogs fully compensate for reduced AMPD1 activity.

Loss-of-function mutation:

The authors should clearly indicate whether c.C34T carriers also carry the K287I variant and, if so, at what frequency. This would clarify any potential linkage between the two variants. Additionally, Figure 4C should be further annotated to include the distribution of c.C34T carriers.

We have now analysed the linkage in full dataset from 1000 Genomes Project. The two variants are in linkage equilibrium ($r^2=0.0004$). Of the 55 chromosomes carrying the K287I variant, not a single one carries the c.C34T variant.

MINOR COMMENTS:

The manuscript states that 8 primate genomes were used for ancestral state inference, but later Figure 1C refers to 13 genomes. This inconsistency needs to be clarified.

We apologize for the oversight and have corrected the figure to refer to 8 primate genomes.

The authors report that the K287I variant is significantly associated with varicose veins in FinnGen, but this association is absent in UK Biobank. The lack of replication is not that surprising due to different approaches, population-specific effects or statistical power issues. However, this issue should be discussed and the authors should weigh in what are the most likely explanations.

We agree that the discrepancy between the FinnGen and UK Biobank results warrants discussion. The reviewer raises several plausible reasons for this inconsistency. We have expanded the Discussion section accordingly.

Reviewer #4 (Remarks to the Author):

We appreciate the time and effort both you and your co-reviewer have dedicated to our manuscript.